# Treatment with Testosterone Therapy in Type 2 Diabetic Hypogonadal Adult Males: A Systematic Review and Meta-Analysis

Kajol Kumari [1], Rohan Kumar [2], Areeba Memon [3], Beena Kumari [3], Moniba Tehrim [4], Pooja Kumari [3], Muhammad Shehryar [5], Hamza Islam [6], Rabia Islam [6], Mahima Khatri [3], Satesh Kumar [7,*] and Ajay Kumar [8]

1. Medicine Department, Ghulam Muhammad Mahar Medical College, Sukkur 65200, Pakistan; kajolkumari919@gmail.com
2. Medicine Department, Jinnah Sindh Medical University, Karachi 75510, Pakistan; dr.rohan234@gmail.com
3. Medicine Department, Dow University of Health Sciences, Karachi 74200, Pakistan; areeba_memon30@hotmail.com (A.M.); tejanibeena29@gmail.com (B.K.); poojatejani583@gmail.com (P.K.); mahimakhatri12333@gmail.com (M.K.)
4. Medicine Department, Karachi Medical and Dental College, Karachi 74700, Pakistan; monibatehrim1@gmail.com
5. Medicine Department, King Edward Medical University, Lahore 54000, Pakistan; shery0334@gmail.com
6. Medicine Department, Punjab Medical College, Faisalabad 38000, Pakistan; hamzaislamdr@gmail.com (H.I.); rabiasindhu99@gmail.com (R.I.)
7. Shaheed Mohtarma Benazir Bhutto Medical College, Lyari General Hospital, Karachi 74200, Pakistan
8. Medicine Department, MedStar Union Memorial Hospital, Baltimore, MD 21218, USA; ajaykumarchhattaani@gmail.com
* Correspondence: kewlanisatish@gmail.com; Tel.: +92-33-2525-2902

**Abstract:** Testosterone replacement therapy (TRT) has been used to treat hypogonadal males with type 2 diabetes mellitus (T2DM) for a long time, despite variable results. This meta-analysis examines TRT's role in hypogonadal males with T2DM. The databases PubMed, Embase, and Google Scholar were searched for relevant RCTs and observational studies. Estimated pooled mean differences (MDs) and relative risks with 95% confidence intervals were used to measure the effects of TRT (CIs). When compared to the placebo, TRT improves glycemic management by significantly reducing glycated hemoglobin (HBA1c) levels (WMD = $-0.29$ [$-0.57$, $-0.02$] $p = 0.04$; I2 = 89.8%). Additionally, it reduces the homeostatic model assessment levels of insulin resistance (WMD = $-1.47$ [$-3.14$, 0.19]; $p = 0.08$; I2 = 56.3%), fasting glucose (WMD = $-0.30$ [$-0.75$, 0.15]; $p = 0.19$; I2 = 84.4%), and fasting insulin (WMD = $-2.95$ [$-8.64$, 2.74]; however, these results are non-significant. On the other hand, HBA1c levels are significantly reduced with TRT; in addition, total testosterone levels significantly increase with testosterone replacement therapy (WMD = 4.51 [2.40, 6.61] $p = 0.0001$; I2 = 96.3%). Based on our results, we hypothesize that TRT can improve glycemic control and hormone levels, as well as lower total cholesterol, triglyceride, and LDL cholesterol levels while raising HDL cholesterol in hypogonadal type 2 diabetes patients. To this end, we recommend TRT for these patients in addition to standard diabetes care.

**Keywords:** hypogonadism; testosterone replacement therapy; TRT; T2DM; type 2 diabetes mellitus; meta-analysis; systematic review; low testosterone levels

## 1. Introduction

Hypogonadism is a clinical syndrome in which the testis fails to produce physiological testosterone levels or a normal percentage of spermatozoa due to pathology in one or more hormonal concentrations of the hypothalamic–pituitary–testicular axis. Male hypogonadism is characterized by low serum total and free testosterone concentrations,

as well as the presence of hypogonadism symptoms [1]. Hypogonadism affects between 5.1% and 12.3% of men aged 30 to 79 years, with an incidence rate of 12.2 per 1000 people annually. Free testosterone levels below 225 pmol/L (65 pg/mL) indicate a pathology and necessitate replacement therapy [2]. Hypogonadism is a global health concern due to its adverse impacts on patients' ability to perform normal activities and overall quality of life. Recent research provides overwhelming evidence linking hypergonadism to type 2 diabetes (T2DM). This is because obesity is associated with an increased risk of testosterone deficiency (TD), which increases fat storage and insulin resistance, and worsens glycemic control [3]. An ever-growing (and at times contradictory) body of research questions whether testosterone should be used in the routine clinical management of type 2 diabetes. Numerous studies have demonstrated that testosterone therapy improves systolic and diastolic blood pressure, lipid profiles, insulin sensitivity, inflammation, fasting plasma glucose (FPG), and glycated hemoglobin (HBA1c) levels in men with type 2 diabetes [4–6]. In addition, long-term testosterone therapy has been proposed to reduce the risk of pre-diabetes developing into type 2 diabetes and to improve the quality of life in men with hypogonadotropic hypergonadism, as measured by the Aging Male Symptoms (AMS) questionnaire [5]. However, there are studies with contradictory findings. Multiple studies have demonstrated that testosterone replacement therapy (TRT) significantly reduces fasting serum glucose (FSG), fasting serum insulin (FSI), and hemoglobin A1C (HBA1C) in hypogonadal patients with type 2 diabetes [6]. In addition, additional data demonstrated that these indicators did not decline significantly in TRT groups [7,8]. Some studies have found a correlation between TRT and lower levels of lipid panel variables, such as total cholesterol (TC), triglycerides (TGs), serum low-density lipoproteins (LDLs), and higher levels of HDLs [7,8]. In contrast, no other studies discovered evidence of a statistically significant increase in lipid metabolism.

The association between TRT and T2DM in male hypogonadism has only been investigated in a few randomized controlled trials and observational studies, with contradictory results. Consequently, we undertook a systematic review and meta-analysis to determine the efficacy profile of TRT in hypogonadal males with type 2 diabetes. This meta-analysis offers the most up-to-date look at how testosterone therapy compares to no treatment or placebo, as far as we are aware.

## 2. Materials and Methods

In accordance with the Preferred Reporting Items for Systematic Review and Meta-Analysis, the Preferred Reporting Items were used to execute this meta-analysis (PRISMA, Berlin, Germany) [9].

PubMed (Medline, State of Maryland, USA) and Cochrane (London, UK) were combed extensively from the study's inception until 25 February 2023. Gray literature and preprints were discovered by searches on ClinicalTrials.gov (Medline, State of Maryland, USA), Google Scholar (California, USA), and Medrxiv (Florida, USA). Medical Subject Headings (MESH terms) and keywords were compiled to form a search strategy. These terms and keywords included ['Testosterone' OR 'TRT' OR Testosterone undecanoate] AND ['Hypogonadism' OR 'Hormonal deficiency'] AND [Diabetes Mellitus']. Information about the search methodology is provided in Table S1. We did not restrict or filter the search results in any way. Manual searches of review articles yielded relevant studies. Titles, abstracts, and body text were assessed by two reviewers in a blinded fashion (MK and SK). Studies of interest were imported into Endnote X9.3.3 (Bld 15659) to prevent duplication (Clarivate Analytics, Philadelphia, PA, USA).

The following criteria were used to select the studies: language, study design, patient population, intervention, comparison, outcomes of interest, and definition.

- English publications.
- Study design: eligible, completed, randomized clinical trials or observational studies were extracted to perform the meta-analysis.

- Patient population: all patients with confirmed type 2 diabetes who met the criteria of hypogonadism.
- Exposure: patients who received testosterone therapy.
- Comparison: this includes the non-TRT group, which received the usual standard of care or placebo.
- Primary outcomes: effects on glucose metabolism and post-treatment hormonal levels.
- Secondary outcomes: cholesterol levels, BMI, waist circumference, body fat, and systolic and diastolic blood pressure.

To ensure the quality of this meta-analysis, the following significant exclusion criteria were established:

- No clear definitions of the diagnosis of late-onset hypogonadism and T2DM, population, dosage and method of testosterone administration, or outcome evaluation.
- Insufficient data for estimating a mean difference (MD) with a 95% confidence interval (CI).
- Duplicates of previous publications.

Furthermore, all included RCTs were evaluated using the 25-item CONSORT checklists, which emphasized describing how trials were conceived, analyzed, and interpreted (Table S2). The number of the 25 reported items was used to assess the quality of the included RCTs. There is a link between the number of reported items and the quality of an RCT. High-quality research includes all 25 criteria.

Two researchers (MK and SK) independently read and analyzed the articles in order to determine their inclusion in the review. Uncertain facts were discussed and clarified. From each trial, we extracted the first author's name, publication year, nation, ethnicity, testosterone cut-off point, diabetes duration, testosterone regimen, medications on comparators, mean age, HBA1c percentage, and total serum testosterone level. These details are condensed in Table 1. Table 2 also includes the measurements for HOMA-IR, fasting plasma glucose, fasting serum insulin, HBA1c, total cholesterol, triglycerides, high-density lipoprotein, low-density lipoprotein, body fat percentage, body mass index, systolic blood pressure, diastolic blood pressure, erectile function, and the aging male score.

The quality of published RCTs was assessed using the modified Cochrane Collaboration risk of bias tool [10], whereas the quality of observational studies was measured using the Newcastle–Ottawa scale [11].

Review Manager 5.4 (Cochrane Collaboration) was utilized to perform the aforementioned meta-analysis. Relative risks (RRs) and 95% confidence intervals (CIs) were calculated for binary outcomes. Continuous outcomes were represented by mean and standard deviation values. This meta-analysis presents the combined effect of relative risks (RRs) and weighted mean differences (WMDs) determined with the generic-inverse variance and continuous outcome functions with a random-effects model. A *p*-value of less than 0.05 was considered statistically significant. Funnel plots were formed for primary outcomes to evaluate potential publication bias.

The amount of discordance between research was estimated using I2 statistics. The I2 value of 25% showed low heterogeneity, the range of 25% to 50% indicated moderate heterogeneity, and the range of 50% and above indicated strong heterogeneity. The effect of individual studies on the overall pooled estimate was examined by conducting a sensitivity analysis on outcomes with a high degree of heterogeneity.

## 3. Results

A total of 659 articles were found in the preliminary literature search. A total of 2 observational [12,13] and 13 randomized trials [5,8,14–24] passed the inclusion criteria for this meta-analysis, reducing the original number of publications to 15. Distinctive characteristics of the included studies are outlined in (Supplementary Tables S2 and S3). The PRISMA diagram illustrates a comprehensive search strategy (Figure 1).

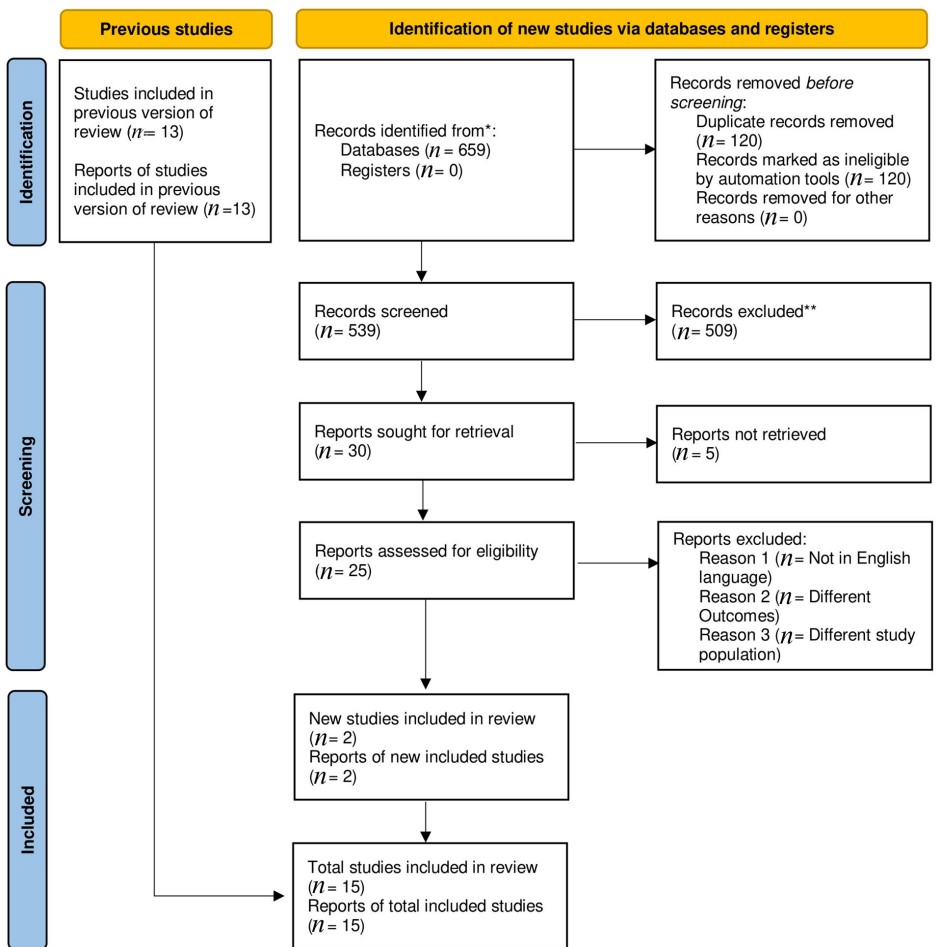

**Figure 1.** Prisma flow chart. * Indicate reporting the number of records identified from each database or register searched (rather than the total number across all databases/registers). ** No automation tools were used.

### 3.1. Baseline Characteristics

There was a total of 3002 participants who fit the criteria for hypogonadism across the 15 studies; 1484 received testosterone and 1518 received a placebo. Hypogonadism was defined as the presence of at least 3 sexual symptoms and a total testosterone level of 12 nmol/L in 6 studies [8,12,13,15,19,21], while in others the presence of a total testosterone level of 15 nmol/L or a free testosterone level of 225 pmol/L was sufficient to diagnose the condition [5,14,16–18,20,22,23]. Yet, another study [14] chose TT13 nmol/L as the threshold for hypogonadism. Included studies used a wide variety of primary testosterone regimens. A total of 1 study [20] utilized oral testosterone, 3 [16,18,22] used testosterone gel by the transdermal method, and 11 used testosterone by a deep intramuscular injection [5,8,12–15,17,19–21,23,24]. Multiple dosing schedules and administration frequencies of testosterone were used in these studies. Of the 13 RCTs, 11 [5,8,14–17,19,21–24] were double-blind placebo-controlled studies, whereas the other 2 [18,20] had no control group at all. Based on the nature of the study, Tables 1 and 2 detail the demographics, health histories, hormone levels, and glycemic indices of the participants.

Table 1. Baseline characteristics of included studies.

| Study | Study Design | Total No. of Patients | Hypogonadism Cut Off Point | No. of Patients | | Age (Mean ± SD) | | Waist Circumference (cm) (Mean ± SD) | | BMI (kg/m$^2$) (Mean ± SD) | |
|---|---|---|---|---|---|---|---|---|---|---|---|
| | | | | TRT | Placebo | TRT | Placebo | TRT | Placebo | TRT | Placebo |
| Dhindsa (2015) [14] | RCT | 34 | FT < 225 pmmol/L | 20 | 14 | 54.6 ± 7.9 | 54.6 ± 7.9 | 128 ± 20 | 124 ± 30 | 39.0 ± 7.6 | 39.4 ± 7.9 |
| Gianatti (2014) [8] | RCT | 67 | TT < 12 nmmol/L | 37 | 30 | 62 ± 2.5 | 62 ± 2.5 | 110 ± 4.2 | 115 ± 2.75 | | |
| Hackett (2014) [15] | RCT | 186 | TT < 12 nmmol/L | 91 | 95 | 61.2 ± 10.5 | 62.0 ± 9.3 | 115.1 ± 13.1 | 112.6 ± 13.3 | 33.0 ± 6.1 | 32.4 ± 5.5 |
| Jones (2011) [16] | RCT | 137 | TT < 11 nmmol/L | 68 | 69 | 59.9 ± 9.1 | 59.9 ± 9.4 | 112.7 ± 13.35 | 111.7 ± 15.23 | 32.76 ± 6.12 | 31.56 ± 5.87 |
| Gopal (2010) [17] | RCT crossover | 22 | FT < 225 pmmol/L | 22 | 22 | 44.23 ± 3.29 | 44.23 ± 3.29 | 93.25 ± 7.03 | 84.10 ± 13.86 | 25.44 ± 3.57 | 22.10 ± 4.93 |
| Heufelder (2009) [18] | RCT | 32 | TT < 11 nmmol/L | 16 | 16 | 57.3 ± 1.4 | 55.9 ± 1.5 | 107.9 ± 1.3 | 105.7 ± 1.4 | 32.1 ± 0.5 | 32.5 ± 0.6 |
| Kapoor (2006) [19] | RCT crossover | 27 | TT < 12 nmmol/L | 24 | 24 | 64 ± 1.34 | 64 ± 1.34 | 115.1 ± 2.4 | 115.1 ± 2.4 | 33 ± 0.86 | 33 ± 0.96 |
| Boyanov (2003) [20] | RCT | 48 | TT < 15 nmmol/ | 24 | 24 | 57.5 ± 4.8 | 57.5 ± 4.8 | N/A | N/A | 31.08 ± 4.79 | 31.01 ± 4.90 |
| Hackett (2018) [21] | RCT | 537 | TT < 12 nmmol/L | 175 | 362 | 58.3 ± 11 | 65.5 ± 11.8 | N/A | N/A | 32.6 ± 6.4 | 31.7 ± 5.9 |
| Yassin (2019) [12] | Observational study | 316 | TT < 12.1 nmol/L | 229 | 87 | 58.2 ± 9.6 | 66.4 ± 7.2 | 104.2 ± 7 | 101.1 ± 9.9 | 30.7 ± 4.1 | 29.8 ± 3 |
| Khirpun (2018) [22] | RCT | 80 | serum levels of total testosterone two times less than 12.1 nmol/L or serum levels of free testosterone two times less than 243 pmol/L in combination with at least two symptoms or complaints of a sexual or psychological nature (total testosterone [TT] below 11 nmol/L and free testosterone below 220 pmol/L) on at least two separate morning measurements after an overnight fast, in addition to exhibiting at least two symptoms of sexual dysfunction (less frequent morning and | 40 | 40 | 53.3 ± 5.4 | 54.1 ± 5.6 | 114.3 ± 9.5 | 114.7 ± 9.8 | 34.0 ± 2.6 | 33.6 ± 2.9 |
| Groti (2020) [5] | RCT | 55 | total testosterone (TT) level <11 nmol/L | 28 | 27 | 58.21 ± 7.94 | 62.19 ± 5.90 | 116.48 ± 5.07 | 115 ± 1.47 | 34.03 ± 4.37 | 32.63 ± 3.67 |
| Groti (2018) [23] | RCT | 55 | and/or free testosterone (FT) level <220 pmol/L | 28 | 27 | N/A | N/A | 116.48 ± 5.07 | 116.64 ± 4.96 | 34.03 ± 4.37 | 32.63 ± 3.67 |
| Wittert (2021) [24] | RCT | 1007 | 13.0 nmol/L | 504 | 503 | 59·8 ± 6·3 | 59.6 ± 6.4 | 118·4 ± 12·6 | 117·8 ± 11·6 | 34·8 (5·1) | 34·6 (5·1) |
| Haider (2020) [13] | Prospective observational | 356 | total testosterone levels ≤12.1 nmol/L (350 ng/dL) and symptoms of hypogonadism | 178 | 178 | 61.5 ± 5.4 | 63.7 ± 4.9 | 116.8 ± 14.3 | 116.9 ± 13.6 | 36.5 ± 4.5 | 33.4 ± 5.3 |

SD: standard deviation, Ft: free testosterone, TT: total testosterone.

**Table 2.** Baseline glucometabolic, lipid, and blood pressure parameters.

| Study | Fasting Plasma Glucose (mmol/L) (Mean ± SD) | | Fasting Serum Insulin pmol/L (Mean ± SD) | | HBA1c % | | Free Testosterone pmol/L (Mean ± SD) | | Total Testosterone nmol/L (Mean ± SD) | | HOMA-IR (Mean ± SD) | | Mean Total Cholesterol mmol/L (Mean ± SD) | | Systolic Blood Pressure mm of Hg (Mean ± SD) | | Diastolic Blood Pressure mm of Hg (Mean ± SD) | |
|---|---|---|---|---|---|---|---|---|---|---|---|---|---|---|---|---|---|---|
| | TRT | Placebo | TRT | Placebo | TRT | Placebo | TRT | Placebo | TRT | Placebo | TRT | Placebo | TRT | Placebo | TRT | Placebo | TRT | Placebo |
| Dhindsa (2015) [14] | 6.99 ± 0.44 | 6.60 ± 0.55 | 13.6 ± 3 rsul | 11.8 ± 2.2 | 6.8 ± 0.9 | 7 ± 1.4 | 156.5 ± 45.11 | 145.74 ± 41.6 | 9 ± 2.9 | 8.3 ± 2.8 | 9 ± 2.9 | 8.3 ± 2.8 | 4.06 ± 0.98 | 4.03 ± 0.95 | N/A | N/A | N/A | N/A |
| Gianatti (2014) [8] | 9.57 ± 3.78 | 9.11 ± 3.65 | N/A | N/A | 7.7 ± 1.3 | 7.5 ± 1.2 | 187.7 ± 57.0 | 181.2 ± 63.6 | 9.2 ± 3.1 | 8.9 ± 3.8 | 4.1 ± 2.0 | 3.7 ± 2.6 | 4.15 ± 0.90 | 4.08 ± 0.9 | 140.2 ± 15.9 | 137.1 ± 13.0 | 79.4 ± 9.4 | 77.5 ± 8.9 |
| Hackett (2014) [15] | 9.05 ± 3.18 | 8.49 ± 2.84 | 20.88 ± 22.83 | 18.17 ± 15.7 | N/A | N/A | 198 ± 49.3 | 202.4 ± 62.1 | 9.2 ± 2.6 | 9.5 ± 3.3 | 5.9 ± 3.8 | 4.9 ± 3.3 | 4.51 ± 1.17 | 4.55 ± 1.01 | 138.6 ± 17.30 | 136.7 ± 17.12 | 82.5 ± 10.23 | 81.6 ± 9.50 |
| Jones (2011) [16] | 7.9 ± 4.3 | 9.2 ± 3.4 | 12.80 ± 8.95 | 17.86 ± 24.72 | 6.43 ± 2.20 | 7.69 ± 2.77 | 177.57 ± 60.19 | 177.57 ± 60.19 | 10.1 ± 3.7 | 10.1 ± 3.7 | 5.50 ± 6.82 | 6.45 ± 8.75 | 4.7 ± 0.9 | 4.0 ± 1.0 | 115.83 ± 5.15 | 118.40 ± 9.97 | 82.00 ± 6.93 | 79.00 ± 3.16 |
| Gopal (2010) [17] | 7.9 ± 0.2 | 8.3 ± 0.2 | 19.03 ± 0.63 | 16.8 ± 0.87 | 7.5 ± 0.1 | 7.5 ± 0.1 | 200 ± 0.00 | 200 ± 0.00 | 10.5 ± 0.2 | 10.4 ± 0.2 | 5.6 ± 0.3 | 6.1 ± 0.4 | N/A | N/A | 104.5 ± 2.6 | 143.5 ± 2.1 | 85.6 ± 0.9 | 85.0 ± 1.0 |
| Heufelder (2009) [18] | 7.83 ± 0.49 | 7.6 ± 0.43 | 13.68 ± 1.95 | 12.37 ± 1.87 | 7.28 ± 0.19 | 7.28 ± 0.19 | N/A | N/A | 8.63 ± 0.51 | 8.63 ± 0.51 | N/A | N/A | 5.11 ± 0.17 | 4.95 ± 0.15 | 127.6 ± 2.8 | 131 ± 3.1 | 74 ± 1.4 | 74 ± 1.4 |
| Kapoor (2006) [19] | 8.0 ± 2.6 | 8.4 ± 2.8 | N/A | N/A | 10.4 ± 1.6 | 10.3 ± 1.6 | N/A | N/A | 9.56 ± 2.33 | 10.76 ± 3.0 | N/A | N/A | 5.50 ± 1.41 | 5.59 ± 1.49 | 122 ± 8 | 120 ± 8 | 80 ± 4 | 76 ± 6 |
| Boyanov (2003) [20] | N/A | N/A | N/A | N/A | 7.6 ± 1.3 | 7.5 ± 1.5 | 210 ± 124.5 | 175 ± 67.9 | 9.7 ± 4.4 | 8.9 ± 3.2 | N/A | N/A | 4.5 ± 1.1 | 4.1 ± 1.0 | 141.8 ± 16.1 | 139.4 ± 16.8 | 81.4 ± 10.4 | 78.2 ± 10.4 |
| Hackett (2018) [21] | 5.3 ± 0.8 | 4.9 ± 1.3 | N/A | N/A | 5.9 ± 0.2 | 5.9 ± 0.2 | N/A | N/A | 8.2 ± 2.1 | 9.6 ± 2.4 | N/A | N/A | 6.9 ± 1.2 | 6.4 ± 1.4 | 136.9 ± 13.5 | 129.8 ± 12.7 | 81.2 ± 8.9 | 84.7 ± 6.7 |
| Yassin (2019) [12] | 8.1 ± 3.7 | 8.7 ± 5.0 | N/A | N/A | 7.8 ± 2.4 | 7.9 ± 2.4 | 208 ± 142 | 223 ± 140 | 9.6 ± 2.7 | 9.9 ± 2.6 | N/A | N/A | 6.1 ± 1.2 | 5.9 ± 1.5 | N/A | N/A | N/A | N/A |
| Khirpun (2018) [22] | 10.06 ± 1.44 | 9.77 ± 1.40 | N/A | N/A | 8.12 ± 1.04 | 7.89 ± 0.77 | 208 ± 142 | 223 ± 140 | 7.24 ± 1.97 | 7.96 ± 1.34 | 11.45 ± 7.34 | 10.82 ± 6.52 | 5.31 ± 0.91 | 5.11 ± 0.85 | 134.64 ± 10.71 | 138.15 ± 13.24 | 77.50 ± 5.85 | 78.89 ± 5.25 |
| Groti (2020) [5] | 10.06 ± 1.44 | 9.60 ± 1.44 | 26.03 ± 15.86 | 24.89 ± 13.90 | 8.12 ± 1.04 | 7.89 ± 0.77 | N/A | N/A | 7.24 ± 1.97 | 7.96 ± 1.34 | 11.45 ± 7.34 | 10.70 ± 6.52 | 5.31 ± 0.91 | 5.31 ± 0.97 | 134.64 ± 10.71 | 138.15 ± 13.24 | 77.50 ± 5.85 | 78.89 ± 5.25 |
| Groti (2018) [23] | 10.06 ± 1.44 | 9.60 ± 1.44 | 26.03 ± 15.86 | 24.89 ± 13.90 | 8.12 ± 1.04 | 7.89 ± 0.77 | N/A | N/A | 7.24 ± 1.97 | 7.96 ± 1.34 | 11.45 ± 7.34 | 10.70 ± 6.52 | 5.31 ± 0.91 | 5.31 ± 0.97 | 134.64 ± 10.71 | 138.15 ± 13.24 | 77.50 ± 5.85 | 78.89 ± 5.25 |
| Wittert (2021) [24] | 6·1 ± 0·9 | 6·1 ± 0·9 | N/A | N/A | 5·7 ± 0·5 | 5·7 ± 0·5 | N/A | N/A | 13·4 ± 4·1 | 13·9 ± 4·6 | N/A | N/A | N/A | N/A | 138.52 ± 14.2 | 139.88 ± 14 | 85.45 ± 8.5 | 85.13 ± 8.3 |
| Haider (2020) [13] | 7.8 ± 1.2 | 6.3 ± 0.7 | 28.6 ± 4.0 | 24.9 ± 2.9 | Forest plot | Forest plot | N/A | N/A | 9.3 ± 1.7 | 9.8 ± 1.1 | 9.8 ± 2.0 | 7.1 ± 1.3 | 8.3 ± 1.1 | 7.1 ± 1.2 | 163.0 ± 13.3 | 145.6 ± 14.6 | 97.6 ± 10.8 | 84.8 ± 10.3 |

SD: standard deviation, HOMA-IR: homeostasis model of insulin resistance, HBA1c: glycated hemoglobin.

### 3.2. Quality Assessment and Publication Bias

The Newcastle–Ottawa scale, a tool used to evaluate the quality of studies, found a minimal likelihood of bias in the observational studies (Supplementary Table S4). Trials of medium to good quality were found using the Cochrane technique of assessing RCTs (Supplementary Table S5). As the funnel plots clearly show, the results are unaffected by publication bias (Supplementary Figure S1).

### 3.3. Primary Outcomes

The effects on glucometabolism (Figure 2).

Testosterone's impact on glucometabolism was evaluated using the HOMA-IR, HBA1c, fasting serum glucose (FSG), and fasting serum insulin (FSI) measures. A total of 9 [5,8,13–15,17,18,22,23] out of 15 studies published data on HOMA-IR, which demonstrated that testosterone therapy reduced HOMA-IR levels more than the placebo (WMD = $-1.47$ [$-3.14$, 0.19]; $p = 0.08$; I2 = 56.3%). Similarly, 14 [5,8,12–20,22–24] of 15 studies measured FSG, and patients in the testosterone group, exhibited a greater decrease in FSG following treatment than those in the placebo group (WMD = $-0.30$ [$-0.75$, 0.15]; $p = 0.19$; I2 = 84.4%). A total of 8 [8,13,14,16–19,23] of 15 studies indicated that post-treatment testosterone patients had a greater reduction in FSI levels (WMD = $-2.95$ [$-8.64$,2.74]; $p = 0.31$; I2 = 49.3%). However, the above-mentioned findings are statistically non-significant. A total of 13 [5,8,12–15,17,18,20–24] of 15 studies provided HBA1c values, and a pooled analysis revealed that testosterone treatment was associated with a significant improvement in post-treatment HBA1c levels (WMD = $-0.29$ [$-0.57$, $-0.02$]; $p = 0.04$; I2 = 89.8%). For illustration, the pre-treatment values for the TRT group were $7.21 \pm 1.09$, while the post-treatment values were $6.60 \pm 1.09$. In contrast, the pre-treatment values for the placebo group were $7.52 \pm 1.08$, whereas the post-treatment values were $7.59 \pm 1.19$.

The effects on hormone levels (Figure 3).

In order to assess the effect of testosterone on hormone levels, the following variables were considered: total testosterone, free testosterone, SHBG, and PSA. A total of 9 studies [5,12–14,19,20,22–24] included total testosterone levels, and the pooled analysis revealed that testosterone therapy is associated with a significant increment in total testosterone levels (WMD = 4.51 [2.40, 6.61] $p = 0.0001$; I2 = 96.3%). Excluding the studies individually from the pooled analysis had no effect on the in-study heterogeneity.

A total of 3 articles [14,15,22] investigated free testosterone levels, and their pooling revealed a significantly greater increase in patients receiving testosterone therapy than the placebo (WMD = 81.21 [23.87, 138.54] $p = 0.07$; I2 = 70%). The SHBG level was included in 5 studies [14,18,22–24], and pooling demonstrated that therapy with testosterone was linked to a greater decrease in SHBG levels (WMD = $-1.28$ [$-5.51$, 2.96] $p = 0.55$; I2 = 0%). A total of 7 studies [8,14–16,18,22,24] assessed PSA levels, and their analysis revealed no clinically significant difference in PSA levels following therapy between the two groups (WMD = $-0.02$ [$-0.13$, 0.08] $p = 0.65$; I2 = 0%).

Secondary outcomes: (Table 3).

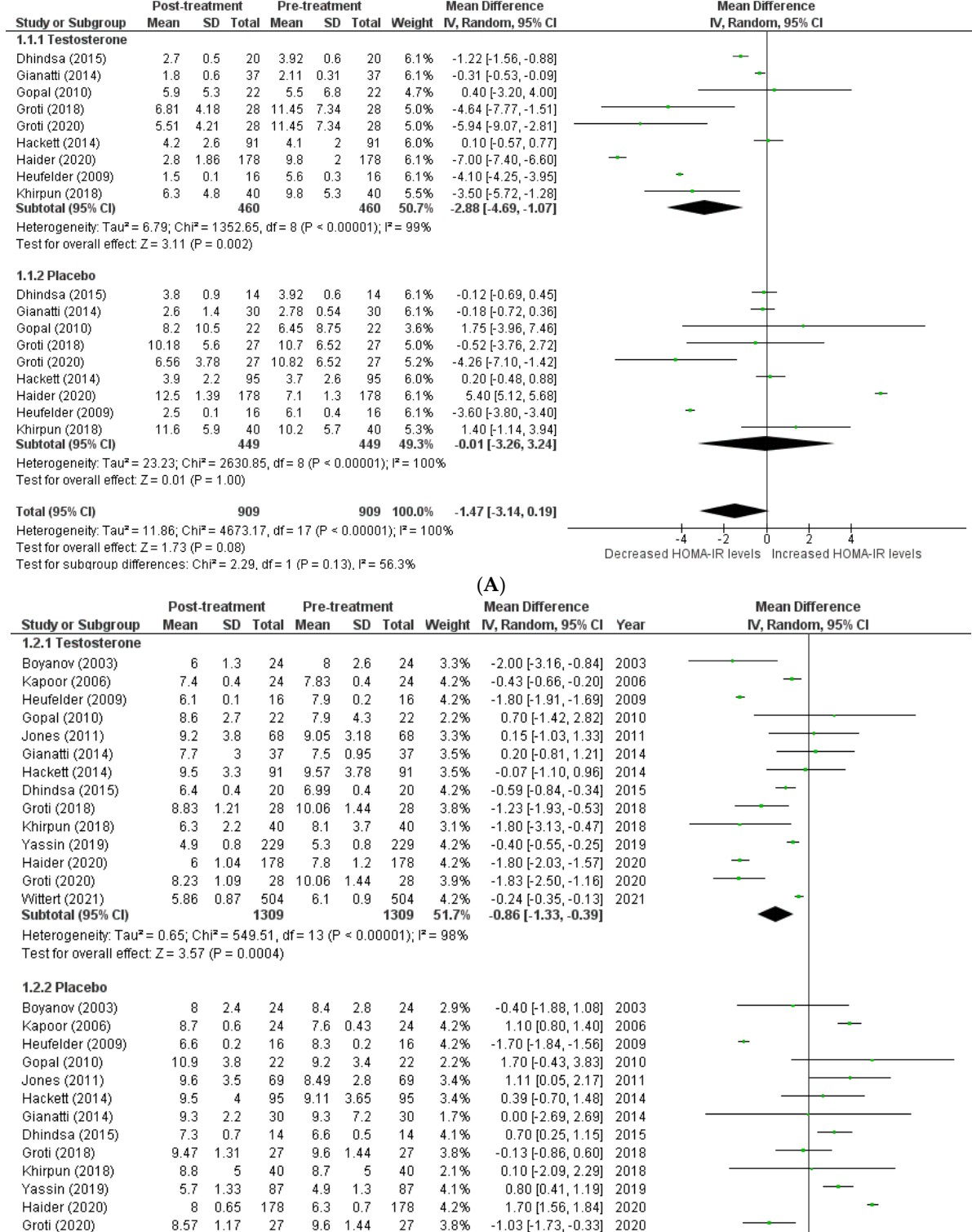

**Figure 2.** *Cont.*

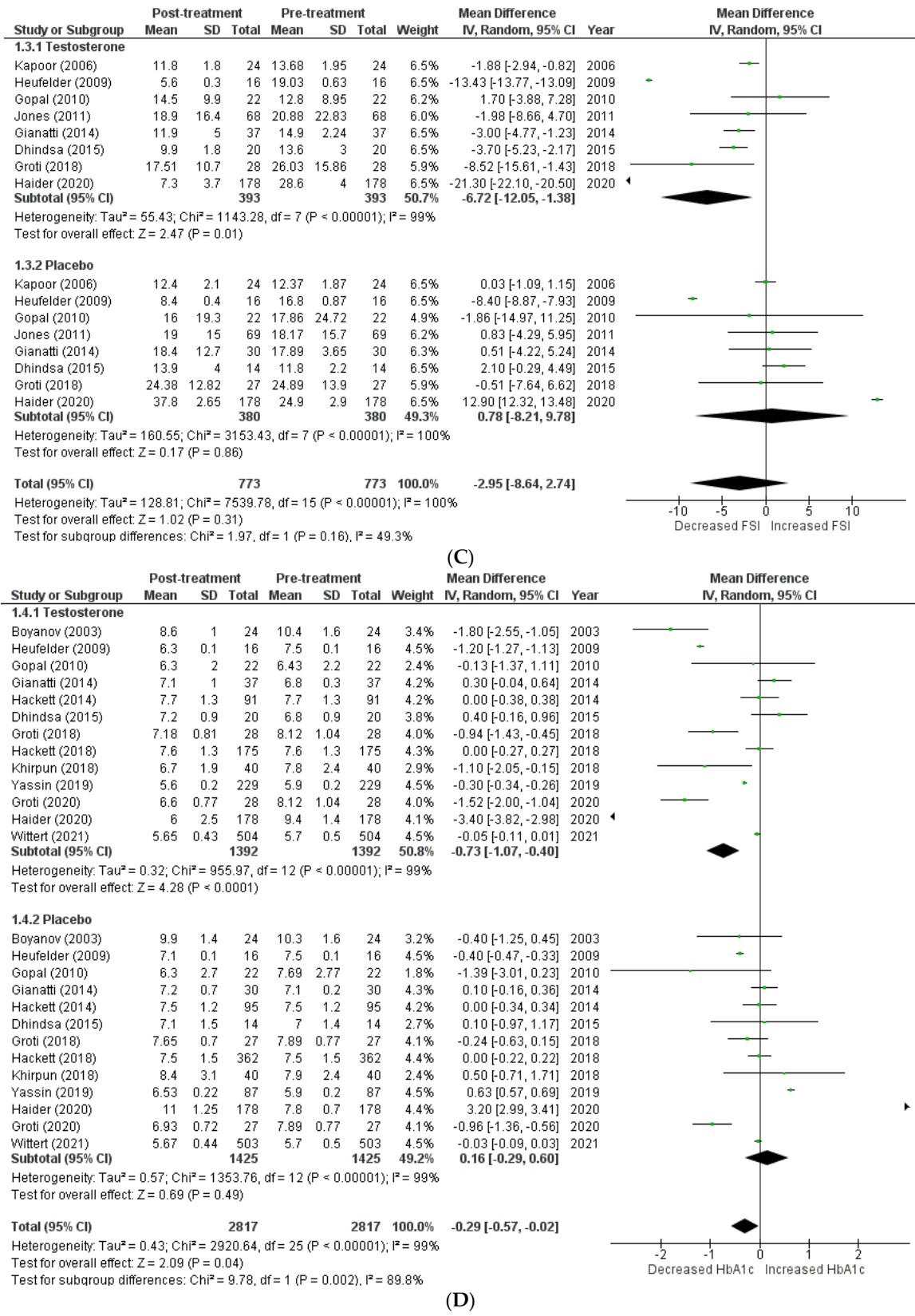

**Figure 2.** Effects on glucometabolism; (**A**) HOMA-IR (homeostatic model assessment for insulin resistance) [5,8,13–15,17,18,22,23]; (**B**) FSG (fasting serum glucose) [5,8,12–20,22–24]; (**C**) FSI (fasting serum insulin) [8,13,14,16–19,23]; (**D**) HBA1c (glycated hemoglobin) [5,8,12–15,17,18,20–24]; WMD—weighted mean difference; CI—confidence interval.

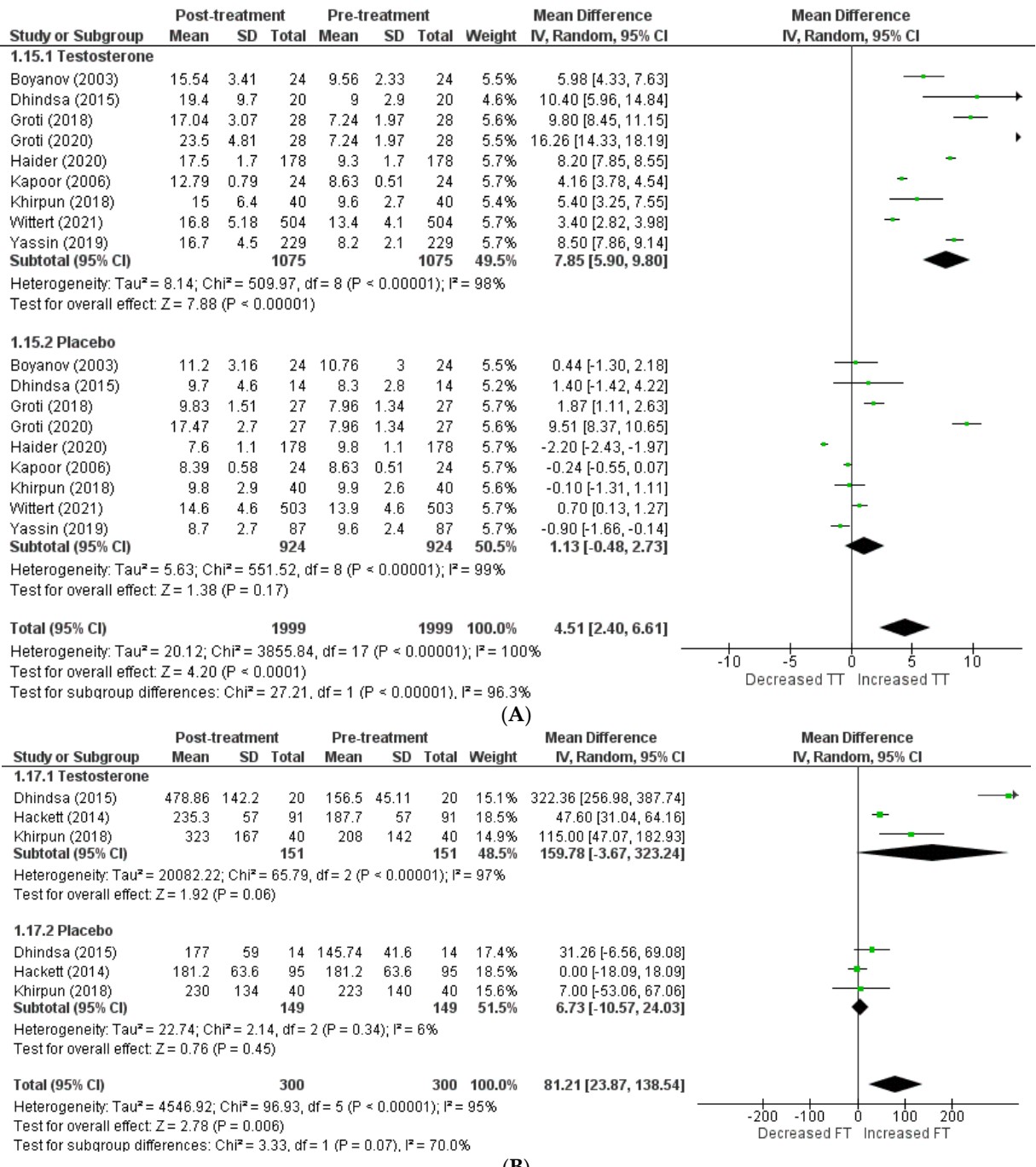

**Figure 3.** *Cont.*

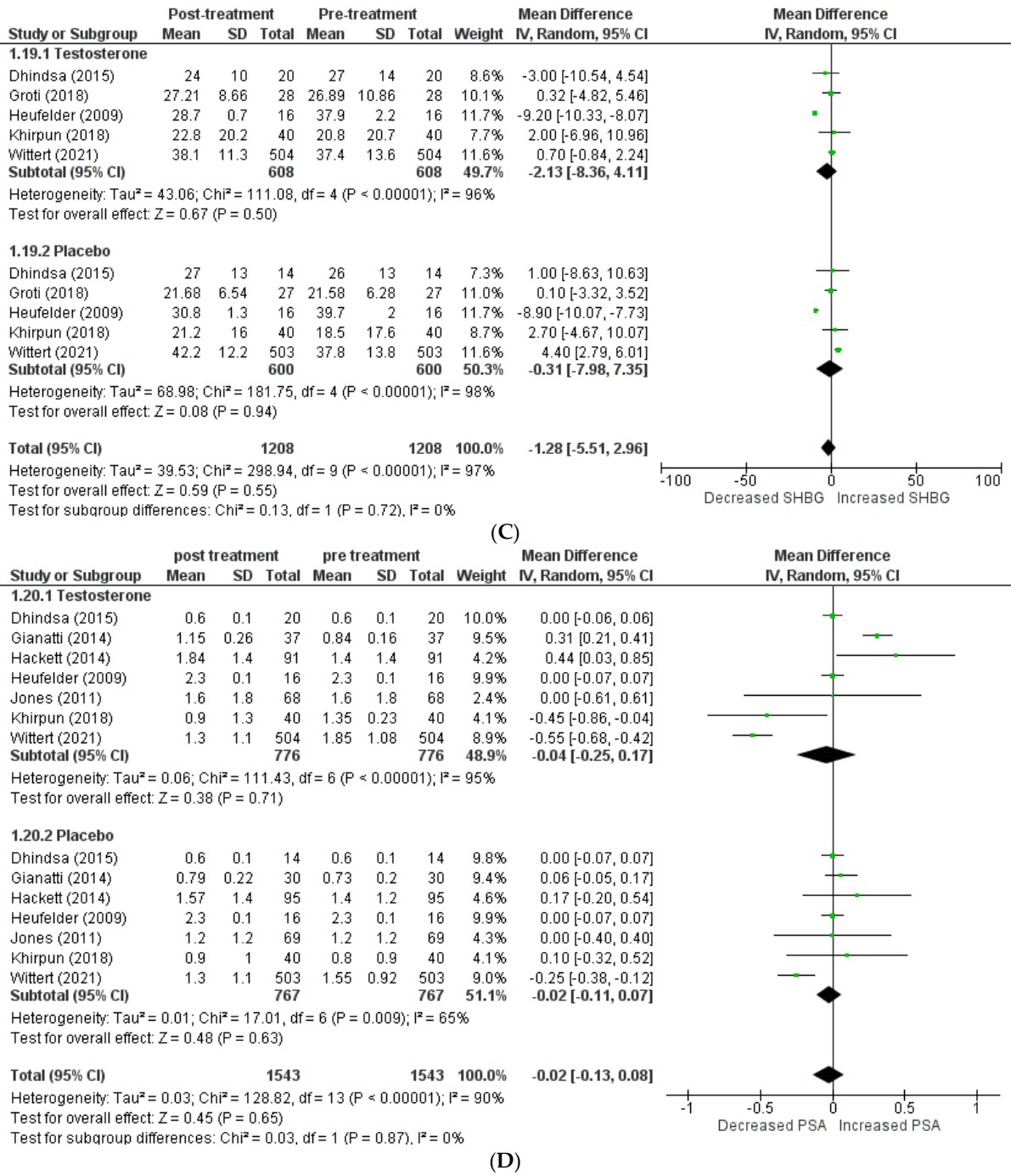

**Figure 3.** Effects on hormone levels, (**A**) TT (total testosterone) [5,12–14,19,20,22–24], (**B**) FT (free testosterone) [14,15,22], (**C**) SBHG (sex−hormone−binding globulin) [14,18,22–24], (**D**) PSA (prostate−specific antigen) [8,14–16,18,22,24].

**Table 3.** Secondary outcomes.

| Outcome | Testosterone | Placebo | Effect Size [CI] | Overall *p*-Value | Heterogeneity |
|---|---|---|---|---|---|
| Total cholesterol | −0.71 [−1.22, −0.21] | 0.10 [−016, 0.35] | −0.32 [−0.64, 0.00] | 0.05 | 87.2 |
| Triglyceride | −0.47 [−0.75, −0.20] | 0.03 [−021, 0.27] | −0.23 [−0.47, 0.00] | 0.05 | 86.6 |
| LDL cholesterol | −0.20 [−1.12, 0.73] | 0.17 [−0.13, 0.46] | −0.02 [−0.52, 0.48] | 0.94 | 0 |
| HDL cholesterol | 0.10 [0.01, 0.20] | 0.03 [−0.07, 0.13] | 0.07 [0.00, 0.13] | 0.04 | 8.7 |
| Body fat | −0.98 [−1.59, −0.38] | −0.54 [−1.12, 0.03] | −0.75 [−1.17, −0.34] | 0.0004 | 6.4 |
| Waist circumference | −3.98 [−6.48, −1.48] | 0.73 [−1.44, 2.89] | −1.68 [3.43, 0.07] | 0.06 | 87.1 |
| BMI | −1.12 [−2.98, 0.74] | 0.05 [−0.51, 0.61] | −0.56 [−1.48, 0.36] | 0.23 | 27.5 |
| SBP | −0.90 [−12.07, 10.26] | −0.19 [−3.19, 2.81] | −0.51 [−6.24, 5.11] | 0.85 | 0 |
| DBP | −3.09 [−5.52, −0.65] | −0.23 [−1.98, 1.52] | −1.68 [−3.16, −0.21] | 0.03 | 71.3 |
| IIEF | 6.98 [3.62, 10.33] | −3.94 [−10.97, 3.10] | 1.66 [−6.75, 10.06] | 0.70 | 86.7 |
| AMS | −16.80 [−26.96, −6.64] | 4.90 [−9.05, 18.85] | −5.94 [−21.87, 9.98] | 0.46 | 83.5 |

CI: confidence interval, LDL: low-density lipoprotein, HDL: high-density lipoprotein, BMI: body mass index, SBP: systolic blood pressure, DBP: diastolic blood pressure, IIEF: international index of erectile function, AMS: aging male score.

In addition to improving HDL cholesterol and IIEF, testosterone treatment has been shown to reduce total cholesterol, LDL cholesterol, triglycerides, body fat, waist circumference, BMI, systolic blood pressure, diastolic blood pressure, and arterial mean stiffness in a pooled analysis of secondary outcomes.

## 4. Discussion

New studies have found that a high percentage of men with type 2 diabetes also suffer from hypogonadism. Despite the increasing knowledge of the association between T2D and hypogonadism, no universally accepted standards exist for patients who suffer from both. This systematic review aimed to develop clear, evidence-based recommendations for treating hypogonadism in males with type 2 diabetes mellitus by administering testosterone replacement therapy. There is solid evidence presenting the link between type 2 diabetes and low blood testosterone levels due to an amplified insulin signaling pathway, as evidenced by multiple studies showing a high incidence (30–80%) of hypogonadism in males with diabetes mellitus [25]. Although several mechanisms mediate hypogonadism, it is more common in males with diabetes than those without diabetes in the Western Hemisphere, Asia, and Africa. This comprehensive systematic review and meta-analysis of 15 studies comprising 3002 patients investigated the effects of testosterone replacement treatment in hypogonadal males with type 2 diabetes compared to the control group (T2DM). The American Academy of Clinical Endocrinologists recommended screening for hypogonadism in all men with type 2 diabetes and all men with a BMI of 30 or a waist circumference of 104 cm in 2016. Even though hypogonadism is common in diseases such as T2DM, the 2018 Endocrine Society guidelines continue to advise against testosterone monitoring [26]. In men with hypogonadism, testosterone replacement therapy (TRT) has the potential to improve sexual desire and function, bone mineral density, muscle mass, body composition, mood, erythropoiesis, cognition, quality of life, and cardiovascular disease; however, the indications for testosterone supplementation are still being debated. The guidelines also divide the potential adverse effects of testosterone replacement therapy into two categories: those with a strong association, such as acne and oily skin, an increase in hematocrit, lower fertility, locally active prostatic carcinoma, and the development of metastatic prostatic carcinoma, and those with a weak association, such as gynecomastia, worsening sleep apnea, and breast cancer progression [27]. Regarding glucometabolic outcomes, our findings show that TRT can greatly enhance glucose control by improving declines in HOMA-IR, FSG, FSI, and glycated hemoglobin (HBA1C), as previously established in other studies [5,8,12,13,20–24]. According to the recent research, baseline HOMA-IR is inextricably linked to BMI, waist circumference, and C-peptide. Furthermore, testosterone supplementation improved metabolic syndrome biomarkers, such as insulin sensitivity, as evidenced by lower HOMA-IR, higher HOMA%, and lower blood C-peptide and proinsulin levels [28]. It has been established that hypogonadal males with diabetes who receive testosterone replacement therapy may see reductions in their body weight and glycemic index values. In a study, researchers discovered that the testosterone treatment

group improved their waist circumference, weight, fasting glucose, HBA1c, blood pressure, lipid profiles, and liver enzymes [29]. In a major testosterone trial involving 788 men over the age of 65 years (at baseline, 72% were obese and 37% had diabetes) with a blood testosterone level of 9.51 mmol/L averaged from two measures, 12 months of testosterone treatment was administered (adjusted to mid–normal concentrations for healthy men) modestly decreased insulin resistance, HOMA-IR 0.6, $p = 0.03$, but had no effect on body weight or waist circumference [29]. Multiple case studies indicate that testosterone therapy is associated with sustained weight loss and a clinically significant reduction in cardiometabolic risk factors, including the complete remission of diabetes. Further interventional studies are needed to fully understand the association between circulating sex hormones and glucose metabolism. Increased peripheral glucose uptake and better insulin sensitivity may result from testosterone's ability to promote the translocation of glucose transporter type 4 (GLUT4) to myocyte membranes. Testosterone's positive effect on glycemic control in the periphery is substantial, even after considering the effects of aging [30].

Our meta-analysis evaluated total cholesterol, high-density lipoprotein (HDL), low-density lipoprotein (LDL), and triglyceride levels as a part of a lipid panel. In thirteen studies, testosterone recipients had lower total cholesterol levels than placebo recipients. Nonetheless, a decrease in triglyceride levels and increased HDL cholesterol were found in 14 trials. However, the two groups had a more negligible difference in LDL cholesterol levels. Similarly, the Si Hyun Kim et al. meta-analysis from 2021 showed that TRT significantly reduced total cholesterol compared to a placebo. Triglyceride levels also improved, albeit the change was not statistically significant. Intriguingly, compared to the placebo, HDL levels significantly decreased after TRT. More contradictory data about TRT's role in HDL are necessary. High-dose TRT has been shown to lower HDL and lipoprotein A levels.

TRT's potential impact on blood lipid and lipoprotein levels is still debated [31]. The effects of testosterone on blood lipid levels are ambiguous. Studies show that men with and without type 2 diabetes have lower HDL levels and higher LDL and triglyceride levels when their testosterone levels are low. According to various cross-sectional studies, there was no connection between raised serum lipid levels or even elevated LDL in patients with high endogenous testosterone profiles [2–4]. In multiple systematic reviews and meta-analyses, it has been demonstrated that TRT dramatically lowers both LDL-C and total cholesterol in men with eugonadal and hypogonadism [32]. People with a high cardiometabolic risk can be screened by measuring their waist circumference and BMI. Testosterone supplementation is gaining acceptance as an anti-obesity drug since it can decrease visceral fat tissue and increase muscle mass in individuals with hypogonadism [32]. Thirteen other investigations, in addition to the planned trials, have shown that testosterone therapy causes a more significant decrease in body mass index [5,8,14–24,32].

Total testosterone, free testosterone, SHBG, and PSA enzyme (kallikrein-3) concentrations were studied. A marked reduction in SHBG accompanied a significant elevation in both total and free testosterone. Nonetheless, no correlation was seen between PSA levels and this treatment. Several meta-analyses have been conducted to date to explore the effect of TRT on PSA [6]. However, the papers investigated did not initially seek to investigate the association between PSA and testosterone, but rather the risk of TRT on the probability of developing prostate cancer. Androgen insufficiency is frequently associated with cardiovascular disease (CVD) risk factors, such as obesity, hypertension, dyslipidemia, and diabetes. PSA is directly regulated by androgen, and several studies suggest that PSA levels may serve as a potential determinant for androgen insufficiency. As Do Kyung Kim et al. demonstrated, TRT dramatically increased PSA levels compared to the placebo [33]. Previous meta-analyses presented a similar significance considering some outcomes; however, their results were non-significant due to smaller sample size, and our sample size was almost double this number [6].

Our meta-analysis provides several advantages: (1) our meta-analysis is strengthened by the inclusion of two additional studies, approximately doubling the total sample size. (2) We also compared our meta-analysis with previous meta-analyses by Jianzhong Zhang et al. [6].

It included eight RCTs as component studies. Two observational studies and five more RCTs, whose publication bias was evaluated using the Cochrane risk of bias tool and the Newcastle–Ottawa Scale, were included in our meta-analysis. (3) A sensitivity analysis was conducted to investigate how different studies affected the overall estimate. (4) Estimates of publication biases were evaluated using a variety of plots and tests, including the funnel plot and Egger's and Begg's tests, and all were shown to be insignificant. The Newcastle–Ottawa Scale was used to look for publication bias in the additional observational study that was included in our meta-analysis. (5) We included the rarely mentioned outcomes of individual studies to account for new information in the literature by integrating the following outcomes: mortality, total testosterone, free testosterone, SHBG, and PSA. (5) Despite having various non-significant results, testosterone therapy for hypogonadism patients with type 2 diabetes can still be relevant and essential for several reasons; it still contributes to the corpus of knowledge on the subject. This is especially crucial if there are few studies available on the topic. This study's findings can assist researchers in identifying any design flaws that may have affected the outcomes. This can aid in the improvement of future study designs. Additionally, it provides data that can be utilized to inform clinical decision-making factors. Clinicians can use this information to assess the possible hazards and advantages of testosterone therapy for their patients. This can also drive future research efforts and ensure that the evidence for testosterone therapy for hypogonadism patients with type 2 diabetes increases. The publication of these data can help prevent this bias and promote more balanced evidence.

Although our study generated significant statistical data, its limitations must be acknowledged. (1) The majority of studies had varying follow-up durations, with some indicating longer times. Longitudinal follow-up studies are preferable when evaluating hormonal diseases, such as hypogonadism, because of the importance of maintaining homeostasis in the body. Numerous studies over a wide range of weeks used testosterone in various doses and administration routes. (2) Variations in study designs, interventions, and patient factors (such as BMI, age, sample size, ethnicity, and trial characteristics) may have contributed to the observed clinical heterogeneity. (3) Only a small number of randomized controlled trials examined the relationship between body fat, AMS and IIEF scores, free testosterone, and death rates; consequently, further research should be conducted in the future, focusing on these outcomes in order to establish a more accurate association. (4) All included RCTs displayed signs of selective reporting bias, except for Groti 2020. Further research was needed to ascertain how testosterone therapy affected libido. (5) Additionally, most studies did not include information on the doses for the control groups, which may have added uncertainty.

### 5. Conclusions

Our results demonstrate that hypogonadal T2DM patients who underwent long-term testosterone replacement therapy experienced a sustained remission of their diabetes. This therapy improved glycemic control, decreased total cholesterol, HDL levels, and triglycerides, and reduced body mass index and waist circumference. We propose that this treatment be taken in conjunction with anti-diabetes medications for these patients. The intervention's long-term durability, safety, and cardiovascular effects need to be studied further.

**Supplementary Materials:** The following supporting information can be downloaded at: https://www.mdpi.com/article/10.3390/clinpract13020041/s1, Supplementary Table S1: Detailed search strategy; Supplementary Table 2: Characteristics of RCTs; Supplementary Table S3: Characteristics of observational studies; Supplementary Table S4: Newcastle–Ottawa scale to assess publication bias in observational studies; Supplementary Table S5: Cochrane risk of bias tool for assessing publication bias in randomized controlled trials; Supplementary Figure S1: Funnel plots of primary outcomes.

**Author Contributions:** Data analysis, manuscript editing, conceptualization, K.K. and R.K.; methodology, A.M.; software, B.K.; validation, M.T., P.K. and M.S.; formal analysis, M.K.; investigation, H.I., R.I.; resources, S.K.; data curation, A.K.; writing—original draft preparation, K.K.; writing—review and editing, R.K.; visualization, M.K.; supervision, S.K.; project administration. All authors have read and agreed to the published version of the manuscript.

**Funding:** This research received no external funding.

**Institutional Review Board Statement:** Not applicable.

**Informed Consent Statement:** Not applicable.

**Data Availability Statement:** No new data were created or analyzed in this study. Data sharing is not applicable to this article.

**Conflicts of Interest:** The authors declare no conflict of interest.

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
