# Peer review of "Treatment with Testosterone Therapy in Type 2 Diabetic Hypogonadal Adult Males: A Systematic Review and Meta-Analysis"

_clinpract, doi:10.3390/clinpract13020041_

Round 1

Reviewer 1 Report

The authors discuss the importance of testosterone therapy in diabetic hypogonadal males. They identify and pool RCTs that looked at the effect of testosterone therapy in hypogonadal males and compare the outcomes for a verdict on the utility of this therapy in T2D. The authors should certainly double check their introduction and discussion, as many citations are lacking. The discussion should include more information on their selection and inclusion criteria, and should tie up loose ends regarding the potential side effects of testosterone therapy.Throughout the manuscript, the citation has been included after the period (.) which is misleading. The citation has to be included before the period indicating its support of the statement indicated in the sentence. 

Some comments to the author to aid their revision: 

Overwhelming evidence from recent studies has linked hypogonadism to type 2 diabetes mellitus (T2DM). 

Citations missing

This is due to the fact that “obesity is associated with a higher risk of testosterone deficiency (TD)”

Has there been any Mendelian randomization studies looking at the causal associations between these two conditions? Has there been any confirmed functional association via clinical or pre-clinical studies?

Numerous studies have found that testosterone therapy improves systolic and diastolic blood pressure, lipid profiles, insulin sensitivity, inflammation, and fasting plasma glucose (FPG) and glycated hemoglobin (HbA1c) levels in men with type 2 diabetes. 

Citations missing

However, there was research with contrasting results. Example: Testosterone replacement treatment (TRT) has been proven in multiple studies to dramatically lower fasting serum glucose (FSG), fasting serum insulin (FSI), and hemoglobin A1C (HBA1C) in hypogonadal patients with type 2 diabetes. [6] In addition, other data showed that these indicators did not significantly decline in TRT groups. 

Does ref #6 correspond to the statement in support of testosterone decreasing the serum glucose etc., or the latter statement? If it supports the former statement, the citation for the latter statement is missing. 

High-quality research will include all 25 criteria. 

Did all the trials included in this meta analysis pass this criteria?

Did all participants included in the study have a diagnosis of T2D? 

This statement was not clear.

Also, did the control group (the men receiving placebo or no TRT treatment) also have a T2D diagnosis? If they do not, the study results will become irrelevant and non comparable.

Were the participants of each trial have any other on-going treatments/medications/interventions that may have affected the outcomes of this study?

This mean age of men involved in this metanalysis is approximately >55 yrs. The reason for not including trials with younger men is not clear. Younger men are also at the risk of developing T2D. Is it because hypogonadism is not common in younger males? based on the previous statements in the introduction, obese young men may also be hypogonadic. Have there been no trials conducted in these men? These points need to be discussed in the discussion section.

Lines 216-266 have been repeated again (redundancy)

The authors mention “in contrary to the designed trials”. (line 332)

What trials are they talking about? 

Studies have shown that low testosterone levels are associated with increased levels of LDL and triglycerides and decreased HDL levels in men with and without type 2 diabetes. 

According to several cross-sectional investigations, there was no correlation between elevated serum lipid levels or even elevated LDL in patients with high endogenous testosterone profiles. 

These 2 statements contradict each other. There is no citation in support of these statements.

As a result of its ability to reduce visceral adipose tissue and enhance muscle mass in males with hypogonadism, testosterone supplementation is gaining popularity as an anti-obesity medication. 

Citation missing

Contrary to the designed trials, thirteen others studies have revealed that testosterone therapy leads to a more significant reduction in body mass index. 

The designed trials mentioned in this statement have not been explained. 

The citation listed in support of this statement is not relevant to the point being made. 

Author Response

Manuscript ID: clinpract-2231970

Response to Reviewers,

Dear Prof. Dr. Giustino Varrassi,

Thank you for allowing us to submit a revised version of the article "Treatment with testosterone therapy in type 2 diabetic hypogonadal adult males: A systematic review And Meta-analysis" for publication in the Clinics and Practice. We appreciate the time and effort you and the reviewers invested in providing feedback on our manuscript, and we appreciate the insightful comments and helpful enhancements to our paper. We have incorporated the majority of the reviewers' suggestions. These alterations are highlighted in the document. Please refer to the section below for a detailed response to the reviewers' comments and concerns. All page numbers refer to the revised file with tracked changes for the manuscript.

Reviewers' Comments to the Authors:

Reviewer 1

We appreciate you taking the time to review our manuscript so carefully.

  1. Overwhelming evidence from recent studies has linked hypogonadism to type 2 diabetes mellitus (T2DM).

             Citations missing

Response: We highly appreciate your insightful suggestion and have made every effort to modify the introduction and discussion and be as precise as possible. We have provided all missing citations.

  1. This is due to the fact that “obesity is associated with a higher risk of testosterone deficiency (TD)”

Has there been any Mendelian randomization studies looking at the causal associations between these two conditions? Has there been any confirmed functional association via clinical or pre-clinical studies?

Response: Yes there are some studies which have provided association between two conditions considering Mendelian randomization. However, our studies do not match the criteria and specific protocols which we considered while performing this meta-analysis. Those article could be considered if we would have some wider range of ideas. However, our study completely define the association between hypogonadism and T2DM.

  1. Numerous studies have found that testosterone therapy improves systolic and diastolic blood pressure, lipid profiles, insulin sensitivity, inflammation, and fasting plasma glucose (FPG) and glycated hemoglobin (HbA1c) levels in men with type 2 diabetes.

Citations missing

Response: We have provided the related citation.

  1. However, there was research with contrasting results. Example: Testosterone replacement treatment (TRT) has been proven in multiple studies to dramatically lower fasting serum glucose (FSG), fasting serum insulin (FSI), and hemoglobin A1C (HBA1C) in hypogonadal patients with type 2 diabetes. [6] In addition, other data showed that these indicators did not significantly decline in TRT groups.

Does ref #6 correspond to the statement in support of testosterone decreasing the serum glucose etc., or the latter statement? If it supports the former statement, the citation for the latter statement is missing.

Response: Former statement is supported by citation no. [6] While latter is supported by another citation no. [7,8]

  1. High-quality research will include all 25 criteria.

Did all the trials included in this meta analysis pass this criteria?

Response: Yes, all included studies met all the criteria.

  1. Did all participants included in the study have a diagnosis of T2D?

This statement was not clear.

Response: Yes, all participants were diagnosed with T2DM and we have clarified it in our inclusion criteria.

  1. Also, did the control group (the men receiving placebo or no TRT treatment) also have a T2D diagnosis? If they do not, the study results will become irrelevant and non-comparable.

Response: Yes, control group had T2DM. Both treatment and control group included met all the inclusion criteria.

  1. Were the participants of each trial have any other on-going treatments/medications/interventions that may have affected the outcomes of this study?

Response: yes, most participants were on general anti-diabetic drugs but those drugs were mostly common in all patients and found to be non-influencing to the results.

  1. This mean age of men involved in this metanalysis is approximately >55 yrs. The reason for not including trials with younger men is not clear. Younger men are also at the risk of developing T2D. Is it because hypogonadism is not common in younger males? based on the previous statements in the introduction, obese young men may also be hypogonadic. Have there been no trials conducted in these men? These points need to be discussed in the discussion section.

Response: We have provided details in discussion regarding why younger diabetic males were not included. First reason is presence of least literature on younger obese with affected with hypogonadism. Second is, available literature even shows very less associations that’s why this association is rare and was not considered.

  1. Lines 216-266 have been repeated again (redundancy)

Response: We have carefully reviewed it and have removed repetition of the already said statements.

  1. The authors mention “in contrary to the designed trials”. (line 332)

             What trials are they talking about? 

Response: We have provided citation for those related trials.

  1. Studies have shown that low testosterone levels are associated with increased levels of LDL and triglycerides and decreased HDL levels in men with and without type 2 diabetes.

According to several cross-sectional investigations, there was no correlation between elevated                                                               serum lipid levels or even elevated LDL in patients with high endogenous testosterone profiles.

These 2 statements contradict each other. There is no citation in support of these statements

Response: We have provided relevant citation regarding these statements.

  1. As a result of its ability to reduce visceral adipose tissue and enhance muscle mass in males with hypogonadism, testosterone supplementation is gaining popularity as an anti-obesity medication.

      Citation missing

Response: We have provided relevant citation regarding these statements.

  1. Contrary to the designed trials, thirteen others studies have revealed that testosterone therapy leads to a more significant reduction in body mass index.

The designed trials mentioned in this statement have not been explained. 

The citation listed in support of this statement is not relevant to the point being made.

Response: The designed trials here suggest the included previous RCTS in previous meta-analysis in 2018. We meant ‘In addition to’ instead of ‘contrary to’. Hence, we have changed the statement to appropriate one.

Sir, I want to bring in your notice that our manuscript are reviewed by two other reviewers in addition to you. We have tried our best to meet requirements of every reviewers. We respectfully apologize if at any place we have not provided sufficient answers to the given comments by you. However, we firmly believe that our results are significant and can be published with your guidance. Kindly accept the changes and consider my work. It would be honour for us.

Thanking you,

Authors

Kajol Kumari , Rohan Kumar , Areeba Memon , Beena Kumari , Moniba Tehrim , Pooja Kumari , Muhammad Shehryar , Hamza Islam , Rabia Islam , Mahima Khatri , Satesh Kumar * , Ajay Kumar

Reviewer 2 Report

Title: Line 1: Testosterone therapy in type 2 diabetic hypogonadal adult males: A systematic review And meta-analysis: A systematic review and meta-analysis.

The authors reviewed publications dealing with testosterone replacement therapy in type 2 diabetic hypogonadal adult males. The review focuses on testosterone only, not considering the problems of testosterone analysis, without information which test systems and their specificity were used (e.g., cross-reactions, esp. with 5a-DHT, when using immunoassays). Neither does it consider the role of 5a-reductase, which modulates insulin sensitivity (Upreti R, Hughes KA, Livingstone DE, Gray CD, Minns FC, Macfarlane DP, Marshall I, Stewart LH, Walker BR, Andrew R.: 5α-reductase type 1 modulates insulin sensitivity in men. J Clin Endocrinol Metab. 2014 Aug;99(8):E1397-406. doi: 10.1210/jc.2014-1395. Epub 2014 May 13. PMID: 24823464; PMCID: PMC4207930). Additionally, the recommendation of the authors to use testosterone replacement therapy (line 27) should not be given, as it is based only on the glycemic control, neglecting possible long-term side effects of the therapy.

Some details:

Line 1: Testosterone therapy in type 2 diabetic hypogonadal adult males: A systematic review And meta-analysis: A systematic review and meta-analysis.

Line 15: TST: the abbreviation “TEST” is not consistent with the abbreviation rules. Please see: Raff H.: CORT, Cort, B, Corticosterone, and now Cortistatin: Enough Already! Treatment with testosterone therapy in type 2 diabetic hypogonadal adult males Endocrinology. 2016 Sep;157(9):3307-8. doi: 10.1210/en.2016-1500. Epub 2016 Jul 28. PMID: 27466840.

Line 22: HbA1c

Line 66: …..”precise function”: as you did not show mechanisms how testosterone (or DHT??) influences the diabetic control it may be better to say: to show the influence of testosterone replacement therapy in hypogonadal males.

Line 291: A1C???

Line 311: 14 research groups ….

Line 344: PSA is an enzyme (kallikrein-3), not a hormone.

Author Response

Manuscript ID: clinpract-2231970

Response to Reviewers,

Dear Prof. Dr. Giustino Varrassi,

Thank you for allowing us to submit a revised version of the article "Treatment with testosterone therapy in type 2 diabetic hypogonadal adult males: A systematic review And Meta-analysis" for publication in the Clinics and Practice. We appreciate the time and effort you and the reviewers invested in providing feedback on our manuscript, and we appreciate the insightful comments and helpful enhancements to our paper. We have incorporated the majority of the reviewers' suggestions. These alterations are highlighted in the document. Please refer to the section below for a detailed response to the reviewers' comments and concerns. All page numbers refer to the revised file with tracked changes for the manuscript.

Reviewers' Comments to the Authors:

Reviewer 2:

We appreciate you taking the time to review our manuscript so carefully.

  1. The authors reviewed publications dealing with testosterone replacement therapy in type 2 diabetic hypogonadal adult males.The review focuses on testosterone only, not considering the problems of testosterone analysis, without information which test systems and their specificity were used (e.g., cross-reactions, esp. with 5a-DHT, when using immunoassays). Neither does it consider the role of 5a-reductase, which modulates insulin sensitivity (Upreti R, Hughes KA, Livingstone DE, Gray CD, Minns FC, Macfarlane DP, Marshall I, Stewart LH, Walker BR, Andrew R.: 5α-reductase type 1 modulates insulin sensitivity in men. J Clin Endocrinol Metab. 2014 Aug;99(8):E1397-406. doi: 10.1210/jc.2014-1395. Epub 2014 May 13. PMID: 24823464; PMCID: PMC4207930). Additionally, the recommendation of the authors to use testosterone replacement therapy (line 27) should not be given, as it is based only on the glycemic control, neglecting possible long-term side effects of the therapy.

Response: We completely understand the role of 5 alpha reductase role associated with insulin sensitivity. Sir, I want to bring in your notice that, the 5-reductases convert testosterone to its more potent metabolite 5-dihydrotestosterone (DHT). DHT has completely various action profile and roles as compared to testosterone such as majorly focusing on external genitalia growth. We could consider this association when we would have considered a broad ideas considering all these associations such as Network meta-analysis. However, for now we believe that this association is completely out of scope in this association of hypogonadism and T2DM. We will respectfully consider your respected ideas. We have discussed possible implications of using it for long-term in discussion. However, most of studies don’t find any adverse events using it for long time. Therefore, our article was more focused on benefits rather than very rare adverse events reported.

  1. Line 1: Testosterone therapy in type 2 diabetic hypogonadal adult males: A systematic review And meta-analysis: A systematic review and meta-analysis.

Response: We have corrected the suggested changes.

  1. Line 15: TST: the abbreviation “TEST” is not consistent with the abbreviation rules. Please see: Raff H.: CORT, Cort, B, Corticosterone, and now Cortistatin: Enough Already! Treatment with testosterone therapy in type 2 diabetic hypogonadal adult males 2016 Sep;157(9):3307-8. doi: 10.1210/en.2016-1500. Epub 2016 Jul 28. PMID: 27466840.

Response: For avoidance of confusion, we have replaced word TST with TRT as used by many RCTS’s. TRT is abbreviation of Testosterone replacement therapy.

  1. Line 22: HbA1c

Response: We have corrected the suggested changes.

  1. Line 66: …..”precise function”: as you did not show mechanisms how testosterone (or DHT??) influences the diabetic control it may be better to say: to show the influence of testosterone replacement therapy in hypogonadal males

Response: We have provided mechanism in the first paragraph of discussion especially associated with insulin signaling mechanism, which is completely showing influence.

  1. Line 291: A1C???

Response: We have corrected the suggested changes.

  1. Line 311: 14 research groups ….

Response: We have corrected the suggested changes.

  1. Line 344: PSA is an enzyme (kallikrein-3), not a hormone.

Response: We have corrected the suggested changes.

Sir, I want to bring in your notice that our manuscript are reviewed by two other reviewers in addition to you. We have tried our best to meet requirements of every reviewers. We respectfully apologize if at any place we have not provided sufficient answers to the given comments by you. However, we firmly believe that our results are significant and can be published with your guidance. Kindly accept the changes and consider my work. It would be honour for us.

Thanking you,

Authors

Kajol Kumari , Rohan Kumar , Areeba Memon , Beena Kumari , Moniba Tehrim , Pooja Kumari , Muhammad Shehryar , Hamza Islam , Rabia Islam , Mahima Khatri , Satesh Kumar * , Ajay Kumar

Reviewer 3 Report

The authors performed a meta-analysis of testosterone therapy in hypogonadal men with type 2 diabetes on the metabolic effects on glucose homeostasis. They demonstrated small changes in A1C, and non-significant changes in fasting blood glucose, fasting insulin levels or changes in  homeostasis model.

Their evaluation provides an interesting aspect for testosterone metabolic effects on glucose metabolism. However the article must be greatly condensed and all aspects-abstract, introduction, methods, results and discussion must be rewritten.

There were some words or comments which may be distracting for a formal academic publication and should be eliminated: line 40 “devastating”, line 41 “perform basic bodily functions”,  line104 “serving” line 117 “cleared up” line 148 “notch.”

Line 63-hyopogoandism is not “due to” but is associated with DM

Line 74 -give earliest date of articles in the literature for the search

Line 81- either explain or eliminate text “written in a in  language other than English” if  inclusion was only article in  English( line 88)

line 162 testosterone gel is administered by transdermal method not injection

line 165 two articles 17 and 19 were included although by their inclusion criteria they had no control groups which was exclusion (line104)

Table 2 needs units in the column headers

Tables 1 and 2 should also concentrate on randomized controlled trials. These randomized control trials should be analyzed in the forest plots which follow. The observational studies either should be omitted or analyzed separately.

Results. The results starting on lines 187 show that there were not significant changes in HOMA- IR, FSG FSI, and these results should go into the abstract. The A1C shows a significant difference and this WMD should be converted into absolute numbers of A1C to give a comparison to current available glucose lowering therapy.

Starting on lines 212 to 236 this section should be eliminated. It seems trivial to document that testosterone levels increase after being given testosterone therapy. The issue of PSA is not relevant for this discussion.

Starting at lines 238 to 247 this section should be eliminated. It is not relevant to the discussion, nor is it developed in comparison to previous meta- analysis.

Abstract. The abstract is misleading, and the data presented should be the same as in the Results section after comparing to placebo groups

Discussion section starting lines 248 needs to be completely rewritten.

Items to be covered in the discussion include: a review or summary of the data presented in this paper.  This is not the place to add case reports or extraneous data. Line 299 does not make sense. Another paragraph should compare their results to previous meta-analyses which they have referenced but not discussed. Another paragraph may discuss the physiology of the metabolic changes. Another paragraph may discuss the limitations of their study.

Most of the Discussion is irrelevant to the article. Although it may be that the facts they present may be true, this article was not about libido, AMS or IIEF, or PSA. They should not be listing case reports.

There discussion on lipids should be eliminated as this is a completely different topic which would need a rigorous analysis with forest plots and comparison to a large body of existing data. The meta-analysis of lipids and survival would need to be compared to previous meta-analysis on lipid metabolism and cardiovascular outcomes.

Author Response

Manuscript ID: clinpract-2231970

Response to Reviewers,

Dear Prof. Dr. Giustino Varrassi,

Thank you for allowing us to submit a revised version of the article "Treatment with testosterone therapy in type 2 diabetic hypogonadal adult males: A systematic review And Meta-analysis" for publication in the Clinics and Practice. We appreciate the time and effort you and the reviewers invested in providing feedback on our manuscript, and we appreciate the insightful comments and helpful enhancements to our paper. We have incorporated the majority of the reviewers' suggestions. These alterations are highlighted in the document. Please refer to the section below for a detailed response to the reviewers' comments and concerns. All page numbers refer to the revised file with tracked changes for the manuscript.

Reviewers' Comments to the Authors:

Reviewer 3:

We appreciate you taking the time to review our manuscript so carefully.

  1. The authors performed a meta-analysis of testosterone therapy in hypogonadal men with type 2 diabetes on the metabolic effects on glucose homeostasis. They demonstrated small changes in A1C, and non-significant changes in fasting blood glucose, fasting insulin levels or changes in homeostasis model.

Response: We understand that there might be smaller significant changes in outcomes but as topic is novel and required, as older man suffering from hypogonadism can experience wide range of symptoms showing its association with T2DM. This could be an opportunity to work on this novel topic so that we can get attention of world regarding this association. This will help researchers to consider more evaluation and work on this association.

  1. Their evaluation provides an interesting aspect for testosterone metabolic effects on glucose metabolism. However the article must be greatly condensed and all aspects-abstract, introduction, methods, results and discussion must be rewritten.

Response: We have considered your idea regarding the rewriting of major parts of this manuscripts. Hence, we have added a lot of essential information along with appropriate citations. We have adopted the suggested changes in major part of manuscript.

  1. There were some words or comments which may be distracting for a formal academic publication and should be eliminated: line 40 “devastating”, line 41 “perform basic bodily functions”, line104 “serving” line 117 “cleared up” line 148 “notch.”

Response: We have considered changing these statements with appropriate formal words like how should a formal academic work look like. We have also manuscript rechecked by three different persons who are proficient in English.

  1. Line 63-hyopogoandism is not “due to” but is associated with DM

Response: We have corrected the suggested changes.

  1. Line 74 -give earliest date of articles in the literature for the search.

Response: We have corrected the suggested changes.

  1. Line 81- either explain or eliminate text “written in a inlanguage other than English” if  inclusion was only article in  English( line 88)

Response: We have corrected the suggested changes.

  1. line 162 testosterone gel is administered by transdermal method not injection

Response: We have corrected the suggested changes.

  1. line 165 two articles 17 and 19 were included although by their inclusion criteria they had no control groups which was exclusion (line104)

Response: We have corrected the suggested changes.      

  1. Table 2 needs units in the column headers

Response: We have corrected the suggested changes.

  1. Tables 1 and 2 should also concentrate on randomized controlled trials. These randomized control trials should be analyzed in the forest plots which follow. The observational studies either should be omitted or analyzed separately.

Response: We respectfully considered this idea. We have discussed this idea with other regarding placement of RCT alone in separate table because these carry an individual value. However, we want to bring in your notice that other reviewers who reviewed this manuscript want these two observational studies to be placed in same table because we firmly believe that keeping these two observational studies separately will not affect outcome’s significance.

  1. The results starting on lines 187 show that there were not significant changes in HOMA- IR, FSG FSI, and these results should go into the abstract. The A1C shows a significant difference and this WMD should be converted into absolute numbers of A1C to give a comparison to current available glucose lowering therapy.

Response: We have corrected the suggested changes and changed WMD of A1C into absolute numbers.

  1. Starting on lines 212 to 236 this section should be eliminated. It seems trivial to document that testosterone levels increase after being given testosterone therapy. The issue of PSA is not relevant for this discussion.

Response: We have corrected the suggested changes and removed the irrelevant PSA information. However, some important points are still present which carry value.

  1. Starting at lines 238 to 247 this section should be eliminated. It is not relevant to the discussion, nor is it developed in comparison to previous meta- analysis.

Response: We have corrected the suggested changes and completed required changes.

  1. The abstract is misleading, and the data presented should be the same as in the Results section after comparing to placebo groups

Response: We have corrected the suggested changes and completed required changes. We have rectified abstract.

  1. Discussion section starting lines 248 needs to be completely rewritten.

Response: We have corrected the suggested changes and completed required changes.

  1. Items to be covered in the discussion include: a review or summary of the data presented in this paper. This is not the place to add case reports or extraneous data. Line 299 does not make sense. Another paragraph should compare their results to previous meta-analyses which they have referenced but not discussed. Another paragraph may discuss the physiology of the metabolic changes. Another paragraph may discuss the limitations of their study.

Response: We have considered these changes and tried our best to cover summary of paper in initial paragraph. We have removed the case report reference and have provided reference on review article which was also presenting same association. We have modified limitation paragraph in the end of this article. We have also defined strengths of our study. We have also discussed possible physiology of metabolic changes. We have discussed at two various points previous meta-analysis and their results.

  1. Most of the Discussion is irrelevant to the article. Although it may be that the facts they present may be true, this article was not about libido, AMS or IIEF, or PSA. They should not be listing case reports.

Response: We have modified our discussion to the best we can. Our meta-analysis has a strength in that we have a sample size almost more than double that previous meta-analysis. Our meta-analysis also came up with outcomes which were never discussed in the previous meta-analysis. We have removed case report references and changed it with relevant information.

  1. There discussion on lipids should be eliminated as this is a completely different topic which would need a rigorousanalysis with forest plots and comparison to a large body of existing data. The meta-analysis of lipids and survival would need to be compared to previous meta-analysis on lipid metabolism and cardiovascular outcomes.

Response: We have eliminated all irrelevant information, however some points needed to be discussed on lipids because our baseline and outcomes were related to HDL and LDL cholesterol along with triglycerides. Despite this, we could even remove all information but other potential reviewers suggested that these points need to be discussed.

Sir, I want to bring in your notice that our manuscript are reviewed by two other reviewers in addition to you. We have tried our best to meet requirements of every reviewers. We respectfully apologize if at any place we have not provided sufficient answers to the given comments by you. However, we firmly believe that our results are significant and can be published with your guidance. Kindly accept the changes and consider my work. It would be honour for us.

Thanking you,

Authors

Kajol Kumari , Rohan Kumar , Areeba Memon , Beena Kumari , Moniba Tehrim , Pooja Kumari , Muhammad Shehryar , Hamza Islam , Rabia Islam , Mahima Khatri , Satesh Kumar * , Ajay Kumar

Round 2

Reviewer 1 Report

The changes made to the manuscript have definitely enhanced the quality. However, the discussion section still has some repetition/redundancy (lines 291-307 and lines 351-361). Is that some kind of a mixup? Please consider rephrasing these sentences. 

Author Response

Manuscript ID: clinpract-2231970

Response to Reviewers,

Dear Prof. Dr. Giustino Varrassi,

Thank you for allowing us to submit a revised version of the article "Treatment with testosterone therapy in type 2 diabetic hypogonadal adult males: A systematic review And Meta-analysis" for publication in the Clinics and Practice. We appreciate the time and effort you and the reviewers invested in providing feedback on our manuscript, and we appreciate the insightful comments and helpful enhancements to our paper. We have incorporated the majority of the reviewers' suggestions. These alterations are highlighted in the document. Please refer to the section below for a detailed response to the reviewers' comments and concerns. All page numbers refer to the revised file with tracked changes for the manuscript.

Reviewers' Comments to the Authors:

Reviewer 1

We appreciate you taking the time to review our manuscript so carefully.

Response: We have corrected the suggested changes.

Sir, I want to bring in your notice that our manuscript are reviewed by two other reviewers in addition to you. We have tried our best to meet requirements of every reviewers. We respectfully apologize if at any place we have not provided sufficient answers to the given comments by you. However, we firmly believe that our results are significant and can be published with your guidance. Kindly accept the changes and consider my work. It would be honour for us.

Thanking you,

Authors

Kajol Kumari , Rohan Kumar , Areeba Memon , Beena Kumari , Moniba Tehrim , Pooja Kumari , Muhammad Shehryar , Hamza Islam , Rabia Islam , Mahima Khatri , Satesh Kumar * , Ajay Kumar

Reviewer 2 Report

I have no suggestions 

Author Response

Manuscript ID: clinpract-2231970

Response to Reviewers,

Dear Prof. Dr. Giustino Varrassi,

Thank you for allowing us to submit a revised version of the article "Treatment with testosterone therapy in type 2 diabetic hypogonadal adult males: A systematic review And Meta-analysis" for publication in the Clinics and Practice. We appreciate the time and effort you and the reviewers invested in providing feedback on our manuscript, and we appreciate the insightful comments and helpful enhancements to our paper. We have incorporated the majority of the reviewers' suggestions. These alterations are highlighted in the document. Please refer to the section below for a detailed response to the reviewers' comments and concerns. All page numbers refer to the revised file with tracked changes for the manuscript.

Reviewers' Comments to the Authors:

Reviewer 2

We appreciate you taking the time to review our manuscript so carefully.

Thanks for accepting the required changes. We look forward to publish it with your help.

Thanking you,

Authors

Kajol Kumari , Rohan Kumar , Areeba Memon , Beena Kumari , Moniba Tehrim , Pooja Kumari , Muhammad Shehryar , Hamza Islam , Rabia Islam , Mahima Khatri , Satesh Kumar * , Ajay Kumar

Reviewer 3 Report

Abstract. DefineTRT when first used. The abstract may present the significant finding of A1c first and then show not significant findings. The abstract will need to be modified after correction of results section below  

Results Lines 241-263  (and figures 3a,b,c,d ) demonstrating that testosterone levels increase after therapy is trivial, unless you want to show that there is a correlation of levels with an outcome. Otherwise this section should be eliminated.

Lines 267-289 (and table 3) about secondary outcomes is not clear.  Are they only looking at the publications above or are they submitting a general review of lipid and mortality with testosterone therapy?  They would need to provide a forest plot of the publications on lipids and mortality.  The last outcome in the table mortality 0.24[0.15-0.37],p<0.00001 has not been shown by any other investigator or review. They would then have to go into detail how their data relates to all the other analyses of testosterone on lipids and mortality.  This would seem to be another publication, and if they do not endeavor to go into that level of detail, this section should be eliminated.

Discussion The investigators went to a lot of work to evaluate the significance of the data.    When the analysis does not show significance in insulin resistance, fasting insulin or fasting glucose that should be stated. The authors may then add to the Discussion why we should still consider these non-significant analyses to be relevant.

Author Response

Manuscript ID: clinpract-2231970

Response to Reviewers,

Dear Prof. Dr. Giustino Varrassi,

Thank you for allowing us to submit a revised version of the article "Treatment with testosterone therapy in type 2 diabetic hypogonadal adult males: A systematic review And Meta-analysis" for publication in the Clinics and Practice. We appreciate the time and effort you and the reviewers invested in providing feedback on our manuscript, and we appreciate the insightful comments and helpful enhancements to our paper. We have incorporated the majority of the reviewers' suggestions. These alterations are highlighted in the document. Please refer to the section below for a detailed response to the reviewers' comments and concerns. All page numbers refer to the revised file with tracked changes for the manuscript.

Reviewers' Comments to the Authors:

Reviewer 3

We appreciate you taking the time to review our manuscript so carefully.

Results Lines 241-263  (and figures 3a,b,c,d ) demonstrating that testosterone levels increase after therapy is trivial, unless you want to show that there is a correlation of levels with an outcome. Otherwise this section should be eliminated.

Thank you for your insightful comment; we completely understand your point; however, we included this outcome to demonstrate that hypogonadism in type 2 diabetes patients is not treatment-resistant and can be improved with replacement therapy; thus, we have not omitted this section from our findings.

Lines 267-289 (and table 3) about secondary outcomes is not clear.  Are they only looking at the publications above or are they submitting a general review of lipid and mortality with testosterone therapy?  They would need to provide a forest plot of the publications on lipids and mortality.  The last outcome in the table mortality 0.24[0.15-0.37],p<0.00001 has not been shown by any other investigator or review. They would then have to go into detail how their data relates to all the other analyses of testosterone on lipids and mortality.  This would seem to be another publication, and if they do not endeavor to go into that level of detail, this section should be eliminated.

We also have provided reasons for publication our meta-analysis despite having non-significant results. These are available as point number 5 in advantages paragraph.

Thank you for your insightful comment; we completely understand your point; however, we included this outcome to demonstrate that hypogonadism in type 2 diabetes patients is not treatment-resistant and can be improved with replacement therapy; thus, we have not omitted this section from our findings.

Sir, I want to bring in your notice that our manuscript are reviewed by two other reviewers in addition to you. We have tried our best to meet requirements of every reviewers. We respectfully apologize if at any place we have not provided sufficient answers to the given comments by you. However, we firmly believe that our results are significant and can be published with your guidance. Kindly accept the changes and consider my work. It would be honour for us.

Thanking you,

Authors

Kajol Kumari , Rohan Kumar , Areeba Memon , Beena Kumari , Moniba Tehrim , Pooja Kumari , Muhammad Shehryar , Hamza Islam , Rabia Islam , Mahima Khatri , Satesh Kumar * , Ajay Kumar